# Positive Effects of Preventive Nutrition Supplement on Anticancer Radiotherapy in Lung Cancer Bearing Mice

**DOI:** 10.3390/cancers12092445

**Published:** 2020-08-28

**Authors:** Yu-Ming Liu, Tsung-Han Wu, Yi-Han Chiu, Hang Wang, Tsung-Lin Li, Simon Hsia, Yi-Lin Chan, Chang-Jer Wu

**Affiliations:** 1Division of Radiation Oncology, Department of Oncology, Taipei Veterans General Hospital, Taipei 11217, Taiwan; ymliu@vghtpe.gov.tw; 2School of Medicine, National Yang Ming University, Taipei 11221, Taiwan; 3Department of Food Science and Center of Excellence for the Oceans, National Taiwan Ocean University, Keelung 20224, Taiwan; u402026@gmail.com; 4Division of Hemato-oncology, Department of Internal Medicine, Chang Gung Memorial Hospital, Keelung 20401, Taiwan; 5College of Medicine, Chang Gung University, Taoyuan 33320, Taiwan; 6Department of Nursing, St. Mary’s Junior College, Yilan 26647, Taiwan; chiuyiham@smc.edu.tw; 7Institute of Long-term Care, Mackay Medical College, New Taipei City 25245, Taiwan; 8Institute of Biomedical Nutrition, Hung-Kuang University, Taichung 43302, Taiwan; sandy720666@gmail.com; 9Genomics Research Center, Academia Sinica, Taipei 11529, Taiwan; tlli@gate.sinica.edu.tw; 10Taiwan Nutraceutical Association, Taipei 104, Taiwan; dr.simon.hsia@gmail.com; 11Department of Life Science, Chinese Culture University, Taipei 11114, Taiwan; 12Department of Medical Research, China Medical University Hospital, China Medical University, Taichung 40402, Taiwan; 13Department of Health and Nutrition Biotechnology, Asia University, Taichung 41354, Taiwan; 14Graduate Institute of Medicine, Kaohsiung Medical University, Kaohsiung 80708, Taiwan

**Keywords:** nutrition, preventive supplement, radiotherapy, lung cancer, sarcopenia, inflammation, EGFR, Bcl-2, VEGF, PD-L1

## Abstract

**Simple Summary:**

The supplement of TNuF, a total nutrition formula, before cancer diagnosis significantly enhanced the anticancer effect of radiotherapy against primary tumor and lung metastasis in tumor-bearing mice. Meanwhile, TNuF improved cancer related sarcopenia. We supposed that these benefits are related to the regulations of cell proliferation, angiogenesis, and immunity in tumor microenvironment.

**Abstract:**

(1) *Background*: Radiotherapy (RT) is one of the major treatments for non-small cell lung cancer, but RT-associated toxicities usually impede its anticancer effect. Nutrient supplementation has been applied for cancer prevention or a complementary measure to anticancer therapy. Here, we explored the influence of total nutrition supplementation before and after cancer occurrence on the anticancer benefit and side effects of RT. (2) *Methods*: C57BL/6JNarl mice were inoculated with Lewis lung carcinoma cells and then treated with radiotherapy. TNuF, a total nutrition formula, was prescribed by oral gavage. In the preventive groups, TNuF supplementation started from seven days before tumor inoculation. In the complementary groups, TNuF supplementation began after tumor inoculation. (3) *Results*: TNuF successfully enhanced the anticancer effect of RT against primary tumor and lung metastasis. Additionally, the complementary supplement improved the high serum TNF-α level and the wasting of sartorius muscle in mice receiving RT. In histologic and molecular analysis, TNuF was observed to modulate EGFR, apoptosis, and VEGF and PD-1/PD-L1 pathways. Furthermore, the anticancer benefit of the preventive supplement was comparable to that of the complementary administration. (4) *Conclusions*: Our results demonstrated that the prescription of the TNuF total nutrition formula before and after cancer diagnosis attains similar benefits in testing subjects with typical anticancer RT. TNuF is also a potential sensitizer to anti-PD-1 immune therapy.

## 1. Introduction

Radiotherapy (RT) is one of the major therapies for patients with non-small cell lung cancer (NSCLC). According to the National Comprehensive Cancer Network (NCCN) guideline, RT is a potential option for all stages of NSCLC, as either palliative or curative treatment [1]. Modern RT uses ionizing radiation to stimulate abundant free radicals surrounding tumor tissue, and hence damages cancer cell DNA by these highly reactive chemicals. Although RT contains an unquestionable benefit to cancer treatment, owing to the destructive effect of free radicals on normal tissues, it also brings patients lots of adverse effects such as anorexia, muscle wasting, suppression of hematopoiesis, and hence influences treatment response and patient survival [2,3]. On the other hand, if a treatment enhances the RT injury to the tumor cell, but not the normal cell, it can magnify the application of RT in NSCLC patients.

The supplementation of natural health products such as coenzyme Q10 (Q10), omega-3 polyunsaturated fatty acid (n-3 PUFA), and selenium plays positive roles in RT and NSCLC. For example, n-3 PUFA has been reported to enhance the cytotoxic effect of RT to colorectal cancer cells through the modulation of inflammatory and apoptotic responses [4]. The pro-oxidant property of the selenium compound is responsible for reactive oxygen species (ROS) generation and cytotoxicity in cancer cells [5]. In an analysis of sixteen clinical studies, selenium supplementation decreased the adverse events rather than the treatment response in patients receiving RT [6]. Meanwhile, the combination of selenium and n-3 PUFA was also found to strengthen antitumor immunity and reduce splenic immunosuppressive cells [7]. Q10 is an endogenous antioxidant, and it can inhibit tumor-associated cytokines [8]. In a clinical case–control study, serum Q10 concentration was reported to be inversely related to the risk of NSCLC [9]. Due to these benefits, many cancer patients have used these healthy products before or after diagnosis. However, there is a lack of research to evaluate the impact of the timing of nutrient uptake to the response of anticancer therapy and the outcome of patients.

TNuF is a total nutrition formula enriched with energy, protein, n-3 PUFA eicosapentanenoic acid (EPA) and docosahexaenoic acid (DHA), Q10, selenium, vitamins, and other micronutrients. TNuF was reported to be able to stabilize body weight, serum albumin, and prealbumin levels in patients undergoing RT [10]. We previously reported that the prescription of TNuF after tumor inoculation significantly enhances the antitumor efficacy of RT, and the combination of TNuF and RT downregulates both epidermal growth factor (EGFR) and vascular endothelial growth factor (VEGF) pathways in tumors of NSCLC-bearing mice [11]. In contrast, the influence of early nutrition supplementation on patients who have not yet been diagnosed with cancer or contracted cancer and are not yet receiving anticancer treatment is rarely studied or discussed. In this paper, we report how the prescription of TNuF before tumor inoculation in mice affects the growth of cancer and the anticancer efficacy of RT. Since the PD-1/PD-L1 pathway is an important route for cancer cells to escape immune surveillance and suppress the antitumor immune reaction [12], we also wanted to know the correlation between the nutrition supplements and tumoral expression of PD-L1.

## 2. Results

### 2.1. Effect of TNuF Combined with Radiotherapy on Tumor Growth

We used tumor-bearing mice to investigate whether the preventive and complementary treatments of TNuF acquire the ability to strengthen the antitumor efficacy of RT. The 5- to 6-week-old C57BL/6JNarl mice were divided equally into seven groups: C group, not receiving tumor inoculation or any treatment; T group, only receiving tumor inoculation; TR group, receiving RT after tumor inoculation; PTN group, receiving TNuF before tumor inoculation; PTRN group, receiving TNuF before tumor inoculation and RT after tumor inoculation; TN group, receiving TNuF after tumor inoculation; and TRN group, receiving the combination treatment of TNuF and RT after tumor inoculation (Figure 1A). Lewis lung carcinoma (LLC) cells were inoculated into the flanks of mice on Day 0. RT was prescribed on Days 8, 10, and 12 after tumor inoculation. The daily prescription of TNuF started on Day -7 (PTN and PTRN groups), Day 0 (TN group), or Day 8 (TRN group) before sacrifice (Figure 1A). We observed the appearance of tumors in all mice seven days after the injection of 3 × 10^5^ cells (Figure 1B). The mean tumor volume achieved 2941 mm^3^ only in T group mice on Day 21. The supplementation of TNuF for both PTRN and TRN groups (before or after tumor inoculation with RT) effectively suppressed cancer growth, and the end-point tumor volume was 497 mm^3^ in the PTRN group and 459 mm^3^ in the TRN groups (both *p* < 0.01 to the T or TR group). When compared to the single treatment with RT or TNuF, the combination treatment of TNuF and RT (PTRN and TRN groups) provides a better inhibitory effect on tumor growth. Next, we resected and weighed the inoculated tumors during sacrifice (Figure 1C,D). The mean tumor weight was 1.0 g in mice of the T group. Single RT (TR group) and the combined TNuF and RT treatment both decreased the tumor weight, where the weight was 0.47 g in the TR group, 0.16 g in the PTRN group, and 0.06 g in the TRN groups (all *p* < 0.05 to the T group). Furthermore, the combination provided a greater reduction than single RT (*p* < 0.05), while the preventive and complementary supplements of TNuF have similar efficacy.

### 2.2. Effect of TNuF on Hematologic and Physiologic Parameters in Irradiated Mouse

Patients with lung cancer frequently manifest in a plunging decrease in body weight, body contents, and blood cells, which may cause poor response to RT and overall survival in patients [13]. To find out whether TNuF could improve these adverse effects in tumor-bearing mice, we measured the murine blood cell counts, lean body weight, skeletal muscles, and fat tissue, together with serum levels of tumor necrosis factor alpha (TNF-α), interleukin 1 beta (IL-1β), and interleukin 6 (IL-6) during sacrifice (Table 1). We found RT (TR group) did not decrease the lean body weight in LLC-inoculated mice, but it attenuated the wasting of sartorius muscle and the elevation of the serum TNF-α level.

The weight of white adipose tissue, the blood cell counts, and serum IL-1β and IL-6 levels were not interfered with by RT. The blood cell counts and hemoglobin in mice receiving tumor inoculation and/or RT were not interfered with by the TNuF supplement either (Table 1). Furthermore, the preventive TNuF supplement (PTN and PTRN groups) was not helpful to alter those cachectic parameters in RT-treated mice, but it did lower down the serum TNF-α level. In contrast, the complementary supplement of TNuF (TN and TRN groups) did not increase the lean body weight, but it effectively attenuated the muscle wasting and the TNF-α level as compared to the no treatment (T) or single RT (TR) group. Interestingly, the complementary TNuF supplement caused a reduction in the white adipose tissue in mice receiving RT. Added together, these results suggested that both TNuF and RT can improve cancer-induced sarcopenia.

### 2.3. Effect of TNuF on Epidermal Growth Factor Receptor Pathway and Mitosis

Given the early studies that the activation of the epidermal growth factor receptor (EGFR) pathway is associated with tumor proliferation and radioresistance in patients [14], we examined the expressions of EGFR, phosphatase and tensin homolog (PTEN), signal transducer and activator of transcription 3 (STAT3), and heat shock protein 90 (HSP90) to evaluate the influence of RT and TNuF (Figure 2). The fluorescence intensity of EGFR in tumor cells was notably reduced by the treatment with RT, TNuF, or the combination (Figure 2A). By contrast, RT, TNuF, and the combination all upregulated the mRNA level of the inhibitor PTEN in tumor cells (Figure 2B). TNuF, RT, and the combination also downregulated the mRNA level of the downstream effector STAT3 (Figure 2C). We did not observe the alternation of the chaperon protein HSP70 in mice receiving TNuF, RT, or their combination (Figure 2D). In addition, the fluorescence intensity of Ki-67 in tumor cells was suppressed in mice receiving these treatments (Figure 2E). Meanwhile, we noticed that the combined TNuF and RT treatments resulted in a stronger inhibition to the EGFR pathway than either single treatment. The preventive supplementation of TNuF induced a higher expression of PTEN than the prescription after tumor inoculation. Added together, these results indicated that the preventive TNuF supplement has an additive effect on RT to modulate tumor mitosis and the EGFR pathway.

### 2.4. Effect of TNuF on Apoptosis in Tumor Cell

Apoptosis is the major cell death modality observed in irradiated tumor cells [15]. Our previous studies also found that the supplementation of n-3 PUFA and selenium can upregulate apoptosis-related proteins in tumor cells [7]. Therefore, we measured the mRNA levels of Bcl-2, Bax, caspase 3, caspase 8, and the chaperon protein HSP70 to follow up the influence of TNuF on cell apoptosis (Figure 3). We observed that a decrease in Bcl-2 mRNA and an increase in caspase 3 mRNA appeared in mice treated with TNuF, RT, or the combination (Figure 3A,D). Within these treatments, the combination with TNuF and RT provided the most effective modulation to tumor Bcl-2 mRNA (Figure 3A). The single supplementation of TNuF was able to upregulate the expressions of Bax and caspase 8 mRNAs (Figure 3B,C). In contrast, the level of HSP70 mRNA was not affected by any treatment (Figure 3E). Taken together, TNuF and RT provided a joint enhancement of tumor cell apoptosis through their modulation on Bcl-2 and caspase 3. In addition, the regulatory function of the preventive TNuF supplement is similar to that of the complementary prescription.

### 2.5. Effect of TNuF on Angiogenic Pathway in Tumor Cell

Tumor angiogenesis is an important predictive factor for RT. The hyperactivation of the angiogenic pathway in tumor cells can trigger a regeneration of tumoral vessels during fractional irradiation, resulting in the failure of RT [16]. In our previous study, the single treatment of TNuF was shown to downregulate the angiogenic pathway in LLC tumor cells [11]. Therefore, we detected the expression of hypoxia-inducible factor 1α (HIF-1α), tyrosine-protein kinase receptor UFO (AXL) mRNA, and vascular endothelial growth factor (VEGF) mRNA to observe the effect of TNuF on the angiogenic pathway in tumor cells (Figure 4). In the IF assay, the fluorescence intensity of tumor HIF-1α was decreased more than 50% in mice treated with RT, TNuF, or the combination (Figure 4A). The complementary TNuF and RT treatment (TRN group) provided a better inhibition of HIF-1α expression as compared to that of the single TNuF supplement (TN group). The mRNA levels of AXL and VEGF were also reduced more than 40% by RT, TNuF, or the combination (Figure 4B). We noticed that TNuF enhanced the inhibitory functions of RT on the expression of VEGF mRNA (Figure 4C). Regarding the timing of the TNuF supplement, the preventive supplement had better inhibition on VEGF mRNA than that of the complementary supplement. Taken together, these phenomena indicated that TNuF has an additive effect on RT to downregulate the tumoral angiogenic pathway.

### 2.6. Effect of TNuF on Immune Response in Non-Irradiated Metastatic Tumor

Several lines of evidence represented that RT not only shrinks the targeted tumor but also results in the shrinkage of non-irradiated tumors elsewhere in the body, and this phenomenon was called the “abscopal effect” [17]. The abscopal effect is related to the RT-induced cell death, the release of tumor antigen, and the activation of tumor-specific CD8+ T cells [18]. Hence, we observed the conditions of tumor metastasis and the expression of T cells within tumor tissues, to evaluate the influence of TNuF on the abscopal effect (Figure 5). In histopathological examination, the tumor load of lung was decreased in mice treated with RT, TNuF or the combination (Figure 5A). Within these treatments, the combination in the PTRN and TRN groups provided the most suppression of lung metastasis. In the IF assay, the fluorescence intensities of CD4 and CD8 were altered and the ratio of CD4+/CD8+ cell was increased in lung tissue after LLC inoculation (Figure 5C). The supplementation of TNuF decreased the expression of CD4+ T cells, increased the infiltration of CD8+ T cells, and therefore downregulated the CD4+/CD8+ ratio in both primary and metastatic lung tumors (Figure 5B,C). In contrast, RT did not have a significant influence on the tumor-infiltrating T cells. Compared to the single treatment and the complementary supplement of TNuF, the preventive TNuF supplement with RT (PTRN group) provided the greatest restoration of the CD4+/CD8+ ratio in tumor tissues (Figure 5C).

Since the upregulation of programmed death-ligand 1 (PD-L1) on the cancer surface can impair the antitumor efficacy of CD8+ T cells [19], we measured the level of PD-L1 mRNA to assess the influence of RT, TNuF, or the combination on the PD-L1 expression (Figure 5D,E). The level of PD-L1 mRNA was downregulated in primary tumors after a treatment with RT or TNuF (Figure 5D). In the lung tissue, RT and TNuF also effectively suppressed the PD-L1 expression of tumor cells (Figure 5E). Further, the inhibitory ability was similar among RT, TNuF, and the combination. Added together, these results point out that the TNuF supplement can enhance the infiltration of CD8+ T cells into primary and metastatic tumors, and therefore it can enhance the abscopal effect of RT.

## 3. Discussion

Natural healthy products, such as n-3 PUFA and vitamins, are believed to have abilities to prevent many diseases from occurring. Many nutrients, such as dietary fiber, fish oils, carotenoids, vitamins, calcium, zinc, and selenium are considered to have abilities to lower cancer risk [20]. This information leads to more and more people intaking healthy products for the prevention of cancer. However, there are few studies of the influence of nutrient uptake on the outcome of cancer patients. In this experiment, we found that the preventive supplementation of nutrients is beneficial to the hosts even after contracting cancer. The mice starting the TNuF supplement before LLC inoculation (PTN and PTRN groups) had smaller primary tumors and milder lung metastasis as compared to the no treatment (T) and single RT (TR) groups. The combination with RT and the preventive TNuF supplement (PTRN group) showed the addictive efficacy by inhibiting tumor growth and metastasis. Meanwhile, the anticancer efficacy of the preventive TNuF supplement (PTN and PTRN groups) was similar to that of the supplement after tumor inoculation (TN and TRN groups). Therefore, the early intake of TNuF before tumor inoculation impairs tumor growth but enhances the efficacy of radiotherapy. Meanwhile, the anticancer benefits were comparable to those in mice receiving TNuF after tumor inoculation. These findings also suggested that cancer patients who had used the nutrient supplement before diagnosis may have a better response than anticancer therapy. Further clinical research is needed to ascertain the positive influence of earlier nutrient intake on cancer patients.

Cancer-related sarcopenia can be induced by cancer itself or anticancer therapy, and it is a negative predictive and prognostic factor to anticancer treatment and cancer patients [20,21]. TNF-alpha is a pro-inflammatory cytokine, and its overexpression is related to sarcopenia and cachexia in lung cancer patients [22]. In this experiment, we first observed that the serum TNF-α level is inversely correlated with the weight of sartorius muscle (*p* = 0.004) but positively related to the lean body weight (*p* = 0.04) and white adipose tissue (*p* = 0.02) in LLC-inoculated mice. Second, the administration of TNuF enhanced the alleviating efficacy of RT on skeletal muscle and circulating TNF-α, but deteriorated the loss of lean body weight and white adipose tissue. Since we did not find any correlation between serum IL-β, IL-6, and these physiologic parameters, circulating TNF-α was supposed to be related to skeletal muscle wasting via promoting tissue proteolysis [23]. The cause and influence of progressive weight loss and lipolysis in TNuF-treated mice were undetermined, which are worthy of further investigation. Although RT has been reported to induce some adverse effects such as muscle wasting in patients of lung cancer [24], RT was shown to improve cancer-induced sarcopenia in this experiment. We propose this controversial effect can be attributed to a short course of RT that turns out to be a good anticancer response to LLC cells.

Although we found that TNuF can downregulate tumoral expressions of caspase 3, Bcl-2, EGFR, VEGF, and HIF-α in our previous research [11], its effect on other pro-tumoral markers, such as STAT3, Ki-67, and AXL, was not evaluated. STAT3 is closely related to the cell proliferation induced by EGFR activation [25]. The proliferation marker Ki-67 is associated with poor response to anticancer treatment and poor outcome in lung cancer patients [26]. In LLC cells, high expressions of Ki-67 and EGFR have been mentioned in the literature [27]. In our experiments, the supplementation of TNuF inhibited EGFR, the downstream signaling protein STAT3, and the mitosis marker Ki-67, and upregulated the inhibitor PTEN, but did not influence chaperon protein HSP90 in LLC tumor cells. Among the components of TNuF, n-3 PUFA has been proven to decrease the phosphorylation of EGFR in human breast cancer cells [28]. 1,25(OH)_2_ vitamin D_3_, the active form of vitamin D, is also shown to upregulate PTEN expression in gastric cancer cell lines [29]. These findings indicate that TNuF can inhibit the EGFR pathway and cell proliferation through modulating EGFR and PTEN proteins.

The conventional RT has been correlated with angiogenesis in the tumor microenvironment. During the fractionated and prolonged exposure of radiation, hypoxic cancer cells release master switch HIF-1α to upregulate expressions of pro-angiogenic factors AXL and VEGF [30,31]. HIF-1α is usually expressed in tumor cells in lung cancer tissues [32]. AXL is an angiogenetic marker and associated with the invasion and progression of lung cancer [33]. HIF-1α also plays a role in the modulation of PD-1/PD-L1, and therefore it provokes cancer resistance to RT and immunotherapy [34]. Some nutrient components of TNuF have been known to be capable of modulating HIF-1α. For example, n-3 PUFA can modulate HIF-1α expression in human cells [35]. The selenium-containing protein Se-methylselenocysteine can inhibit HIF-1α, HIF-2α, and VEGF in clear cell renal carcinoma [36]. The supplementation of TNuF is shown to inhibit HIF-1α and its downstream proteins AXL and VEGF in this study, indicating TNuF has an extra functionality on anti-angiogenesis. By means of blocking tumor HIF-1α expression, TNuF can effectively enhance the anticancer response of RT.

Yokouchi, H. et al. and Chamoto, K. et al. had reported an increase in tumor-specific CD8+ T cells in LLC-bearing mice under radiotherapy [37,38]. In our experiments, we observed that the ratio of CD4+/CD8+ T cells and the expression of PD-L1 in primary and metastatic tumors were all suppressed by the TNuF supplement, which are correlated with the influence of TNuF on the abscopal effect of RT. The insufficiency of nutrients, such as Q10, n-3 PUFA, selenium, and vitamins, has also been correlated to some immunodeficiency diseases and cancers in men [39,40]. Selenium and PUFA were shown to activate anticancer cytotoxic T cells and inhibit pro-cancer regulatory T (Treg) cells and myeloid-derived suppressor cells (MDSCs) in cancer hosts [7]. Since the abscopal effect is related to the release of tumor antigen and the infiltration of CD8+ cytotoxic T cells in tumor tissues after RT [41], the overall anticancer immune response can thus be attributed to the additive effect of TNuF and RT.

The mRNA levels of PD-L1 in primary and metastatic tumors were suppressed by RT or TNuF in our experiments. This result seems to be contradictory to the perception that RT upregulates PD-L1 through activating the EGFR signaling pathway [42]. In the literature, the upregulation of PD-L1 in tumor cells is correlated with two mechanisms—the extrinsic pathway which depends on interferon gamma produced by NK and CD8+ T cells, and the intrinsic pathway which relies on the EGFR/JAK2 signaling pathway [42,43]. Hence, the contradictory conclusion may be due to differences in cell line selection (LLC in this study) and study design (animal versus the supplementation) to those in the literature (e.g., using glioma or head and neck cancer cell lines under given experimental conditions). Nevertheless, the activation of programmed cell death protein-1 (PD-1) and PD-L1 in cancer cells has been linked to the suppression of tumor antigen-specific T cells and disease progression [41]. Although RT may decrease the PD-L1 expression in tumor tissues by directly killing cancer cells, the prolonged exposure of irradiation is linked to the upregulation of PD-L1, thus leading to treatment failure [44]. Therefore, the TNuF supplement is conducive to lessen the tumor resistance to RT by its inhibition of PD-L1. Further experiments are needed to elucidate the effect of TNuF on EGFR and PD-L1 expressions in glioma or head and neck cancer cells in the context of the same animal system.

The supplementation of TNuF was found to affect the apoptosis of tumor cells. The impact of TNuF was presented to upregulate Bcl-2, the inhibitor of the intrinsic pathway, and to inhibit caspase 3, the downstream signaling activated by both intrinsic and extrinsic pathways. The upregulation of Bcl-2 is one of the predominant ways for cancer cells to evade apoptosis [45]. Since TNuF did not steadily modulate the expressions of Bax, caspase 8, and HSP70, the major impact of TNuF is considered as an inhibiter against Bcl-2. Of the components of TNuF, 1,25(OH)_2_ vitamin D_3_ has been proven to induce a caspase-independent apoptosis in breast cancer cells through inhibition of Bcl-2 [46]. Selenium was shown to induce downregulation of Bcl-2 and Ki-67 expressions in many cancer cell lines [5]. n-3 PUFA also appeared to suppress Bcl-2 but upregulate p53, caspase 3, and caspase 7 in Walker 256 cancer cells [47]. Taken together, our results converge on one fact that TNuF can induce apoptosis of cancer cells through suppressing Bcl-2-related proteins.

Selenium and n-3 PUFA have been considered as important regulators for cell death and intracellular signaling in tumor cells. Husbeck et al. and Fico et al. reported that selenium administration can induce selective cytotoxicity in adenocarcinoma cell lines, but not in normal cells [48,49]. Additionally, n-3 PUFA has been proven to cause selective cytotoxicity against multiple cancer cell lines but no significant toxicity against normal cells [50]. These findings indicate that selenium and n-3 PUFA, two major components of TNuF, exhibit cell-specific toxicity on cancer cells rather than normal cells. Schley et al. noted that n-3 PUFA can decrease the phosphorylation of both EGFR and p38 MAPK in cancer cells [28]. Chen et al. stated that selenium can decrease EGFR expression in tumor cells and cancer metastasis [51]. Nair et al. reported that methylselenic acid and selenite can decrease PD-L1 and VEGF expressions in cancer cells [52]. Given lung cancer cells which are resistant to EGFR tyrosine kinase inhibitors, Liao et al. discovered that the combination of selenium and n-3 PUFA was able to reverse the acquired resistance in cancer cells [53]. Therefore, we propose that selenium and n-3 PUFA are two major components involved in the TNuF-induced intracellular signaling and growth inhibition in LLC cells. In future experiments, we will continue to explore the regulations of critical intracellular signaling molecules and growth inhibition in LLC cells treated with selenium and n-3 PUFA.

Despite that taking nutrients is now highly prevalent, most reports remain focused on their usage after cancer diagnosis. The general benefits of nutrition supplements after cancer diagnosis include the cooperative effects with anticancer RT, the activation of tumor apoptosis, the downregulation of EGFR and VEGF pathways in cancer cells, and the attenuation of cancer-induced sarcopenia [4,6,10,11,13]. By contrast, we found that the preventive supplementation of TNuF acquires similar benefits as administration after tumor inoculation. On top of that, if the nutrients can be kept at a sufficient level in blood or tumor tissues, the nutrients may exert a beneficial early intervention effect for both cancer prevention and treatment.

There are some limitations and weaknesses in this experiment. For example, we only used one line of lung cancer cells to assess the effects of nutrition intake, which may not represent the whole subtypes of non-small cell lung cancer. Although Western blot (WB) and quantitative real-time polymerase chain reaction (RT-qPCR) are standard ways to quantify protein expressions in some fields, the immunofluorescence method may have met the general qualitative description for the current study. Concerning the evaluation of signaling proteins, we did not discuss the influence of the treatment on the post-translational regulation level because these issues are better to be addressed in depth as a separate study.

In summary, TNuF, a total nutrition formula including n-3 PUFA, selenium, coenzyme Q10, and other micronutrients, is an effective radiosensitizer to inhibit tumor growth and distant metastasis. The addictive ability of TNuF may be a consequence of a net positive modulation of EGFR, apoptosis, VEGF, and PD-1/PD-L1 in tumors. TNuF also reduces cancer-associated sarcopenia via the suppression of the serum TNF-α level. Importantly, the benefits of the preventive TNuF administration are no less than the supplementation given after contracting cancer. These results demonstrated that the benefits of long-term nutrient uptake are in relation to positive treatment response and outcome for cancer patients. Given that TNuF is a potential PD-L1 inhibitor, we plan to examine the effects of TNuF plus immune checkpoint inhibitors on cancer cells in our next studies.

## 4. Materials and Methods

### 4.1. Cell Line and Culture

In this manuscript, we want to describe the synergic effect of radiotherapy and our total nutrition formula to lung cancer-bearing mice. Since we had confirmed the anticancer effect of this nutrition regimen on many lung cancer cell lines in the previous experiments [7,11], we only used the Lewis lung carcinoma (LLC) cell line in this animal experiment. LLC cells were purchased from the Bioresource Collection and Research Center (BCRC, Hsinchu, Taiwan). Cells were maintained according to the BCRC instructions. In brief, LLC cells were cultured in Dulbecco’s modified Eagle’s medium (DMEM, Sigma-Aldrich, St. Louise, MI, USA) supplemented with 10% fetal bovine serum (FBS, Sigma-Aldrich), 2 mmol/L L-glutamine, 100 U/mL penicillin, and 100 mg/mL streptomycin at 37 ℃ in a humified incubator containing 5% CO_2_.

### 4.2. Animal Tumor Model

Male C57BL/6JNarl mice at age of 5-6 weeks were purchased from the National Laboratory Animal Center (NLAC, Taipei, Taiwan). Mice were housed in a climate-controlled room with a 12:12 dark–light cycle and a constant room temperature of 21 ± 1 ℃. Mice adapted themselves to a new environment and diet at least 4 days before starting experiments. All experiments were approved by the Institutional Animal Care and Use Committee (IACUC) at National Taiwan Ocean University (NTOU), and the experiments conformed to the protocol IACUC-105020 approved by the IACUC ethics committee of NTOU.

In the experiments, mice were divided into seven weight-matched groups (*n* = 6 per group, Figure 1A): (1) C group: control receiving normal saline; (2) T group: LLC-inoculated mice receiving normal saline; (3) TR group: LLC-inoculated mice treated with RT; (4) PTN group: mice receiving TNuF from 7 days before LLC inoculation; (5) PTRN group: mice receiving TNuF from 7 days before LLC inoculation and treated with RT after tumor inoculation; (6) TN group: LLC-inoculated mice receiving TNuF administration; (7) TRN group: LLC-inoculated mice treated with TNuF and RT. Here, 3 × 105 LLC cells in 100 μL normal saline were inoculated into the right posterior flank of each C57BL/6JNarl mouse on Day 0. TNuF powder, at the dose of 1 g in 5 mL normal saline, was prescribed every day via oral gavage from either Day -7 (PTN and PTRN groups), Day 0 (TN group), or Day 8 (TRN group) until sacrifice. Tumor volume [V = L (longest diameter) × W^2^ (shortest diameter) × 0.5] was measured once every two days after LLC inoculation. On Day 21, mice were anesthetized and sacrificed. Following sacrifice, murine tumors, lungs, gonadal white adipose tissues, and blood were collected and stored for further analysis. The condition of lung metastasis was inspected visually after sacrifice, and the visible tumors were dissected for the following experiments. Lean body weight on Day 21 was measured by the following equation: [Lean body weight = Murine body weight–Inoculated tumor weight].

### 4.3. Components of the TNuF

TNuF powder is an energy-dense and protein-rich oral nutrition formula. It is a commercially available product (tradename: NUTRAWELL^Ⓡ^) manufactured by DoWell Laboratories (Tustin, CA, USA) and provided by New Health Enterprise, Inc. (Irvine, CA, USA). Each gram of TNuF powder contains 3.95 Kcal of energy, 51 mg of carbohydrate, 23 mg of protein, 3.6 mg of eicosapentanenoic acid (EPA), 2.4 mg of docosahexaenoic acid (DHA), and multiple micronutrients such as vitamins, selenium, and Q10. Table 2 shows the compositions of the TNuF powder.

### 4.4. Radiotherapy

Mice of the TR, PTRN, and TRN groups received RT with 3 Gy per fraction, one fraction per day on Days 8, 10, and 12 (Figure 1A). The RT field encircled the gross tumor and a 3–5 mm margin at the right posterior flank of the mouse. A 10 MeV photon beam was delivered by a linear accelerator (Clinac 2100C; Varian Associates, CA, USA). The dose rate was 2.4 Gy per minute. Full electron equilibrium was ensured for each fraction using a parallel-plate ionization chamber (PR-60C; Capintel, NJ, USA).

### 4.5. Blood Sample Preparation and Analysis

A total of 0.5 mL blood for each mouse was collected during sacrifice. The complete blood cell counts and leukocyte differential counts (CBC/DC) were calculated by an automated analyzer (Symex K-1000; Sysmex American, IL, USA). Serum albumin level was measured by a dry chemical system (SPOTCHEMEZ SP-4430; Arkray, Japan). Serum levels of TNF-α and IL-1β were determined by ELISA (Quantikine ELISA Kit; R&D system, MN, USA) and a spectrophotometer (mQuant; Bio-Tek, VT, USA). All assays were performed following the protocols provided by the manufacturers.

### 4.6. Immunofluorescence Assay

Tissue presentations of Ki-67, CD4, CD8, EGFR, and HIF-1α were measured by immunofluorescence (IF) assay. Sections were prepared and blocked using a blocking buffer solution for 1 h at room temperature and then incubated by a staining buffer solution containing the primary antibody at 1:200 dilution for 24 h. The extra primary antibody was washed away by PBS. Sections were stained by a staining buffer solution containing the secondary antibody at 1:100 dilution for 24 h at room temperature and then washed with PBS. The primary antibodies were used as follows: anti-mouse Ki-67 (Cat # MA5-14520; Thermo Fisher, MA, USA), anti-mouse CD4 (Cat # 14-0042-82; Thermo Fisher, MA, USA), anti-mouse CD8 (Cat # MA1-145; Thermo Fisher, MA, USA), anti-mouse EGFR (Cat # RM-2111; Thermo Fisher, MA, USA), and anti-mouse HIF-1α (Cat # bs-0737R; Bios, MA, USA). The secondary antibody was anti-rabbit IgG FITC conjugated (Cat # A120-100F; Bethyl Laboratories, TX, USA). The blocking buffer solution contained 1× PBS, 5% normal serum, and 0.3% Triton X-100. The antibody dilution buffer solution contained 1× PBS, 1% BSA, and 0.3% Triton X-100. Relative quantification was performed using ImageJ software (NIH, MD, USA).

### 4.7. RNA Extraction and Real-Time PCR

The RNeasy Mini Kit (Cat # 74104; Qiagen, MD, USA) was applied to extract RNA from inoculated or lung tumors, while the M-MLV Reverse Transcriptase (Cat # M1701; Promega, WI, USA) and Oligo(dT)15 Primer (Cat # C1101; Promega, WI, USA) were used to synthesize cDNA. Real-time PCR was performed using the iCycler iQ System (Bio-Rad, CA, USA). Quantitative real-time PCA analysis was carried out in a 25-μL reaction consisting of 12.5 μL iQ SYBR Green Supermix (Cat # 1708880; Bio-Rad, CA, USA), 5 μL cDNA, RNase-free water, and 100 μM of each primer. Values were normalized to GAPDH mRNA amount. The oligonucleotide primers for mouse AXL (5′-GGTGTTTGAGCCAACCGTGGAA-3′ and 5′-GCCACCTTATGCCGATCTACCA-3′), mouse Bax (5′-TGCTACAGGGTTTCATCCAG-3′ and 5′-GTCCAGTTCATCTCCAATTCG-3′), mouse Bcl-2 (5′-CCTCACCAGCCTCCTCAC-3′ and 5′-ACTACCTGCGTTCTCCTCTC-3′ and 5′-ACTACCTGCGTTCTCCTCTC-3′), mouse caspase 3 (5′-GGAGATGGCTTGCCAGAAGA-3′ and 5′-ATTCCGTTGCCACCTTCCT-3′), mouse caspase 8 (5′-ATGGCTACGGTGAAGAACTGCG-3′ and 5′-TAGTTCACGCCAGTCAGGATGC-3′), mouse HSP70 (5′-CAGCGAGGCTGACAAGAAGAA-3′ and 5′-GGAGATGACCTCCTGGCACT-3′), mouse HSP90 (5′-CCTGAAGGTCATCCGCAAGAAC-3′ and 5′-GGCGTCGGTTAGTGGAATCTTC-3′), mouse PD-L1 (5′-GCTGAAAGTCAATGCCCCATA-3′ and 5′-TCCACGGAAATTCTCTGGTTG-3′), mouse PTEN (5′-CATTGCCTGTGTGTGGTGATA-3′ and 5′-AGGTTTCCTCTGGTCCTGGTA-3′), mouse STAT3 (5′-AGGAGTCTAACAACGGCAGCCT-3′ and 5′-GTGGTACACCTCAGTCTCGAAG-3′), and mouse VEGF (5′-GATGTATCTCTCGCTCTCTC-3′ and 5′-CTTCTCAGGACAAGCTAGTG-3′) were used according to previously published sequences.

### 4.8. Statistical Analysis

All experiments were performed three times, each time in triplicate. Data were expressed as means ± standard deviation (SD). Data were analyzed by Student’s t-test as compared between two groups. A *p* value < 0.05 is considered statistically significant.

## 5. Conclusions

TNuF, a total nutrition formula, has antitumor and anti-proliferative effects on lung cancer cells. In addition, it may be associated with the regulation of angiogenesis and immunity in the tumor microenvironment. No matter when the prescription is provided, either before or after the onset of cancer, the benefits are similar. These results suggest that patients who received adequate and sufficient nutrition before cancer diagnosis bode a good prognosis and treatment response. In addition, TNuF enhances the anticancer efficacy of RT, downregulates the immunosuppressive pathway of cancer cells, and attenuates cancer-related sarcopenia. This information indicates TNuF is a potential sensitizer in anti-PD-1 immunotherapy.

## Figures and Tables

**Figure 1 cancers-12-02445-f001:**
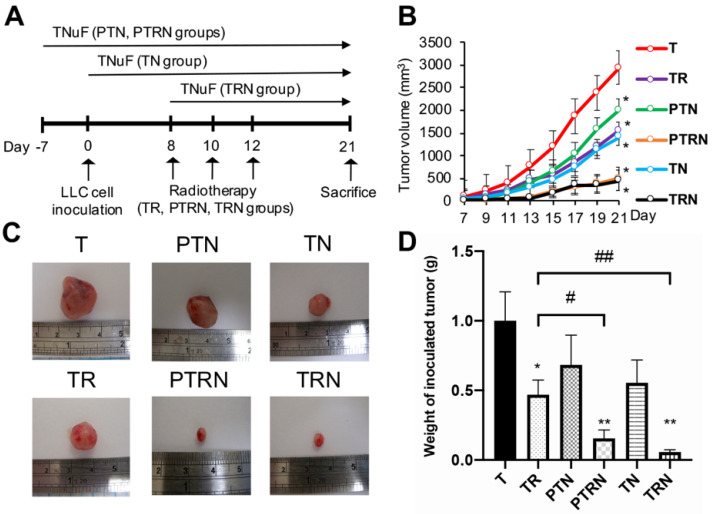
Total nutrition formula enhances anticancer effect and inhibits cell proliferation in tumors receiving radiotherapy. (**A**) Each group of mice was composed of six male C57BL/6JNarl mice. Here, 3 × 10^5^ Lewis lung carcinoma (LLC) cells were inoculated into the right posterior flank region of C57BL/6JNarl mice on Day 0, except for mice of the C group, which received subcutaneous injection of normal saline instead. The TNuF total nutrition formula, 1 g/mice/day, was prescribed orally (PO) from Day -7 in the PTN and PTNR groups, from Day 0 in the TN group, and from Day 8 in the TRN group. In contrast, mice of the C, T, and TR groups were fed with regular rodent diet. In the TR, PTRN, and TRN groups, mice received external beam radiation (RT) to the right posterior flank tumor at 300 cGy/fraction, one fraction per day on Days 8, 10, and 12. All mice were sacrificed on day 21. (**B**) Tumor volume growth curve of various treatment (*n* = 6). (**C**) Representative inoculated tumors at sacrifice. (**D**) The inoculated tumor weight of each group measured at sacrifice (*n* = 6). Data are expressed as means ± SD. *, *p* < 0.05 as compared to the T group. **, *p* < 0.01 as compared to the T group. #, *p* < 0.05 as compared to the TR group. ##, *p* < 0.01 as compared to the TR group.

**Figure 2 cancers-12-02445-f002:**
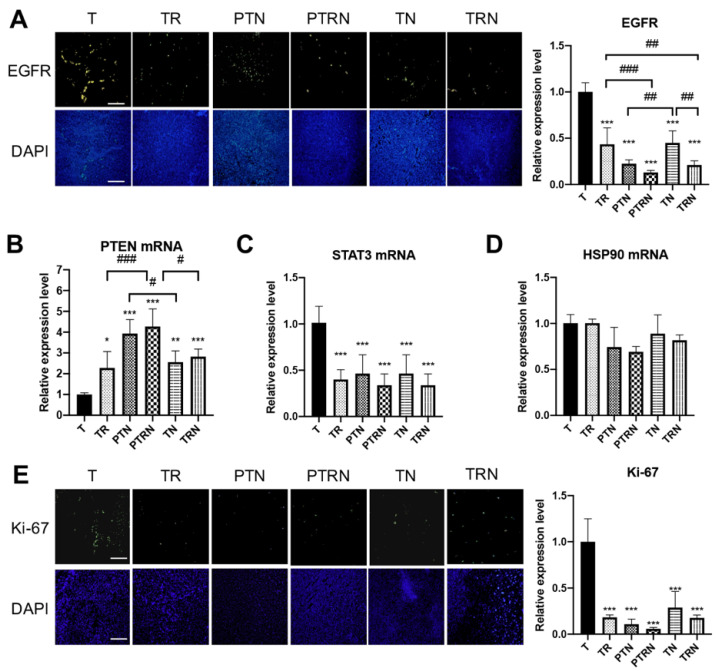
Total nutrition formula downregulates the expression of the epidermal growth factor receptor pathway in tumors receiving radiotherapy. (**A**) Immunofluorescence analysis for epidermal growth factor receptor (EGFR) in inoculated tumor tissues. Scale bar = 200 μm. The relative expression of levels of (**B**) PTEN, (**C**) STAT3, and (**D**) HSP90 were measured by RT-PCR. (**E**) Immunofluorescence analysis for Ki-67 in inoculated tumor tissues. Scale bar = 200 μm. Six tissue sections of each group were selected for immunofluorescence, RT-PCR, and statistical analysis. Data are expressed as means ± SD. *, *p* < 0.05 as compared to the T group. **, *p* < 0.01 as compared to the T group. ***, *p* < 0.001 as compared to the T group. #, *p* < 0.05 as compared between two groups. ##, *p* < 0.01 as compared between two groups. ###, *p* < 0.001 as compared between two groups.

**Figure 3 cancers-12-02445-f003:**
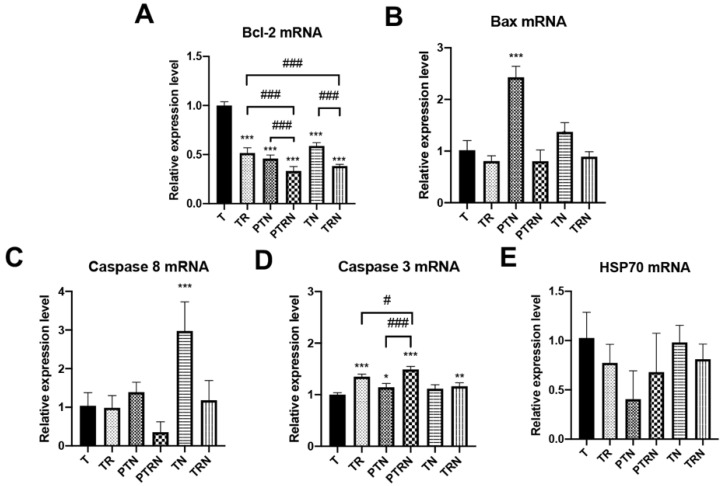
Total nutrition formula upregulates apoptosis in tumors receiving radiotherapy. Relative expression levels of (**A**) Bcl-2, (**B**) Bax, (**C**) caspase 8, (**D**) caspase 3, and (**E**) HSP70 RNAs in the inoculated tumor were measured by RT-PCR (*n* = 6). Data are expressed as means ± SD. *, *p* < 0.05 as compared to the T group. **, *p* < 0.01 as compared to the T group. ***, *p* < 0.001 as compared to the T group. #, *p* < 0.05 as compared between two groups. ###, *p* < 0.001 as compared between two groups.

**Figure 4 cancers-12-02445-f004:**
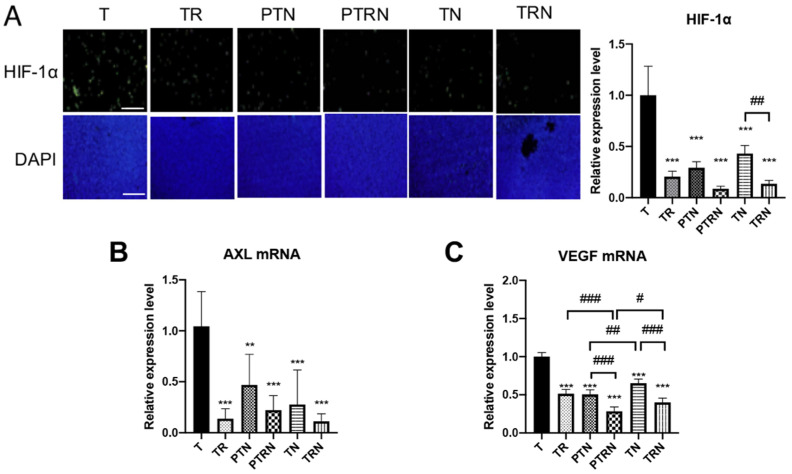
Total nutrition formula suppresses angiogenesis in inoculated tumor tissue receiving radiotherapy. (**A**) Immunofluorescence analysis for hypoxia-inducible factor 1α (HIF-1α) in inoculated tumor tissues. Scale bar = 200 μm. The relative mRNA expression levels of (**B**) tyrosine-protein kinase receptor UFO (AXL) and (**C**) vascular endothelial growth factor (VEGF) were measured by RT-PCR (*n* = 6). Six tissue sections of each group were selected for immunofluorescence analysis and statistical analysis. Data are expressed as means ± SD. **, *p* < 0.01 as compared to the T group. ***, *p* < 0.001 as compared to the T group. #, *p* < 0.05 as compared between two groups. ##, *p* < 0.01 as compared between two groups. ###, *p* < 0.001 as compared between two groups.

**Figure 5 cancers-12-02445-f005:**
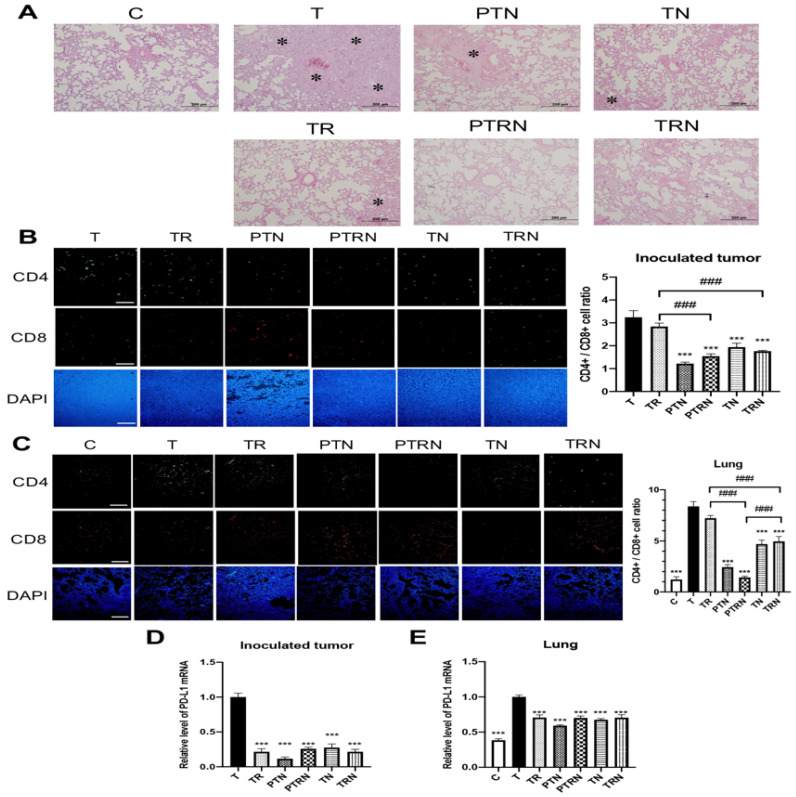
Total nutrition formula enhances the anticancer immunity of radiotherapy. (**A**) Hematoxylin and eosin (H&E) staining of lung tissues. Spots indicate metastatic tumors. Scale bar = 200 μm. (**B**) Immunofluorescence analysis for CD4+ (green) and CD8+ (red) T cells in inoculated tumor tissues. Scale bar = 200 μm. (**C**) Immunofluorescence analysis for CD4+ (green) and CD8+ (red) T cells in lung tissues. Scale bar = 200 μm. (**D**) The relative expression level of PD-L1 mRNA in the inoculated tumor. (**E**) The relative expression level of PD-L1 mRNA in the lung metastatic tumor. Six tissue sections of each group were selected for immunofluorescence analysis and statistical analysis. Data are expressed as means ± SD. ***, *p* < 0.001 as compared to the T group. ###, *p* < 0.001 as compared between two groups.

**Table 1 cancers-12-02445-t001:** Effect of the TNuF total nutrition formula on hematologic and physiologic parameters.

Hematologic Parameters						
Treatment	RBC (10^6^/mm^3^)	Hgb (g/dL)	PLT (10^4^/mm^3^)	WBC (10^3^/mm^3^)	GRAN (10^3^/mm^3^)	LYM (10^3^/mm^3^)
C	10.7 ± 1.0 *^#^	16.7 ± 1.5 *^#^	26.7 ± 13.4	14.4 ± 3.0	2.9 ± 0.9	10.6 ± 2.4 ^#^
T	9.2 ± 0.9	14.3 ± 1.3	35.2 ± 15.5	12.1 ± 4.5	2.4 ± 0.9	8.4 ± 3.4
TR	9.0 ± 0.8	14.4 ± 1.1	22.3 ± 5.8	10.1 ± 1.9	2.5 ± 0.9	6.6 ± 1.1
PTN	9.3 ± 0.7	14.4 ± 1.2	28.8 ± 12.6	13.4 ± 3.2	2.8 ± 1.1	9.4 ± 2.0
PTRN	9.7 ± 1.1	15.5 ± 1.7	22.2 ± 8.4	9.9 ± 2.1	2.1 ± 0.8	7.0 ± 1.3
TN	10.0 ± 0.6	15.5 ± 0.9	25.8 ± 9.1	13.6 ± 2.4	2.6 ± 0.7	9.9 ± 1.6
TRN	9.1 ± 0.6	14.7 ± 1.0	19.9 ± 4.6	9.1 ± 1.7	2.0 ± 0.7	6.4 ± 1.0
**Physiologic Parameters**						
**Treatment**	**Lean BW** **(g)**	**Sartorius M** **(mg)**	**WAT** **(mg)**	**TNF-α** **(pg/mL)**	**IL-1β** **(pg/mL)**	**IL-6** **(pg/mL)**
C	22.2 ± 1.5 ^#^	6.2 ± 0.8 *	27.2 ± 3.5 *^#^	0.5 ± 0.2 *^#^	4.8 ± 0.5	2.2 ± 0.2
T	21.1 ± 1.6	3.9 ± 1.3 ^#^	22.2 ± 2.7	11.6 ± 1.7 ^#^	4.8 ± 0.4	2.8 ± 0.8
TR	20.2 ± 1.3	5.4 ± 0.9 *	22.4 ± 3.7	8.5 ± 0.8 *	4.3 ± 0.1	2.6 ± 0.5
PTN	19.8 ± 1.0	4.5 ± 0.9	20.1 ± 4.5	6.3 ± 1.4 *	4.1 ± 0.1 *	2.5 ± 0.4
PTRN	19.2 ± 0.7 *	5.2 ± 0.6	19.1 ± 5.3	5.7 ± 1.2 *^#^	4.7 ± 0.1	2.5 ± 0.4
TN	19.3 ± 1.1 *	5.9 ± 0.7 *	17.1 ± 3.2 *^#^	6.8 ± 0.7 *	4.3 ± 0.2	2.4 ± 0.4
TRN	20.3 ± 0.4	6.2 ± 0.9 *	17.1 ± 2.3 *^#^	5.3 ± 0.7 *^#^	4.8 ± 0.2	2.4 ± 0.5

*n* = 6 in each group. C, control. T, tumor inoculation. TR, tumor inoculation + radiotherapy. PTN, TNuF before tumor inoculation. PTRN, TNuF before tumor inoculation + radiotherapy. TN, TNuF after tumor inoculation. TRN, TNuF + radiotherapy after tumor inoculation. RBC, red blood cell. Hgb, hemoglobin. PLT, platelet. WBC, white blood cell. GRAN, granulocyte. LYM, lymphocyte. Lean BW, lean body weight. Sartorius M, sartorius muscle. WAT, white adipose tissue. *, *p* < 0.05 as compared to the T group. #, *p* < 0.05 as compared to the TR group.

**Table 2 cancers-12-02445-t002:** Major components of the TNuF total nutrition formula.

Component (Unit)	Amount per 100 g TNuF Powder
Total calories (Kcal)	395
Total carbohydrate (mg)	5100
Total protein (mg)	2300
Total fat (mg)	1200
Eicosapentaenoic acid (mg)	360
Docosahexaenoic acid (mg)	240
Cholesterol (mg)	0
Retinyl acetate (IU)	1067
β-Carotene (IU)	1067
Vitamin C (mg)	133
Vitamin D (IU)	133
Vitamin E (mg)	40
Vitamin K (mcg)	67
Vitamin B1 (mg)	2
Riboflavin (mg)	1.6
Niacin (mg)	16
Vitamin B6 (mg)	2
Folate (μg)	223
Vitamin B12 (μg)	4.4
Biotin (μg)	67
Choline (mg)	27
Sodium (mg)	453
Calcium (mg)	467
Iron (mg)	6
Phosphorous (mg)	267
Iodine (μg)	80
Magnesium (mg)	120
Zinc (mg)	11
Selenium (μg)	80
Copper (μg)	333
Manganese (mg)	2.4
Chromium (μg)	120
Molybdenum (μg)	73
Potassium (mg)	733
Coenzyme Q10 (mg)	40

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
