# Peer review of "Positive Effects of Preventive Nutrition Supplement on Anticancer Radiotherapy in Lung Cancer Bearing Mice"

_cancers, 2020, doi:10.3390/cancers12092445_

Round 1

Reviewer 1 Report

Major comments:

  1. The authors have conducted a similar research previously to evaluate the combination effect of TNuF and radiotherapy as well as their possible mechanisms using ame animal model and materials (Nutrients. 2019, 11(8): 1944). The results of this published article indicated that TNuF can inhibit tumor metastasis and to enhance radiotherapy effect through activating apoptosis pathway (casapase-3 and Bcl-2) and suppressing the expressions of tumor progression-associated molecules (EGFR, VEGF and HIF-α). Thereby, the novelty is less in the present study, and more advanced data should be provided to substantially improve the quality of this study.
  2. The authors indicated that the single treatment with RT or TNuF did not significantly affect the growth of tumors (Lines 88-89). However, all groups except T group have been labeled with a statistical asterisk at day 21 (Fig 1Β).
  3. The authors stated that RT will result in sarcopenia. However, the intervention of TR does not change lean BW as well as it can largely increase sartorius M as compared to that of the T group (Table 1). is this result implied that the animal tumor model used in this study cannot truly reflect clinical findings or phenomenon? Moreover, the authors also stated that TNuF can improve RT-induced sarcopenia. However, no significant difference has been found between TR and intervention groups of TNuF (PTRN and TRN; Table 1).
  4. No description regarding hematologic parameters (Table 1) has been mentioned in the paragraph of Results.
  5. The authors stated that combination of TNuF and TR has synergistic effect in many regulations. However, are there any results to differentiate or evidence this phenomenon is a synergistic or additive effect. Maybe, “combination index” should be examined.
  6. All pictures have poor resolution, and their size are too small. It’s difficult to recognize or extract any useful information from them.
  7. Main purpose of immunohistostaining analysis is to understand the locations or distributions of detected proteins within tissue sections. This technique is not suitable for quantifying expression levels of detected proteins. One of reasons is detected proteins are usually heterogeneously distributed within a tissue. Thereby, it’s hard to estimate the overall level of a protein within a tissue based on a small amount of data obtained from some tissue sections. Maybe, Western blot or RT-qPCR would seem a more appropriate choice to quantify expression levels of detected molecules.
  8. How do the authors confirmed that detected signals are distributed in tumor cells, vascular cells or extracellular matrix on tissue sections?
  9. Proteins play critical effectors in transductions and activations of intracellular signaling pathways, whose processes not only involved in change of expression level but also post-translational regulations (e.g. protein phosphorylation, degradation from latent form to active form) and translocations. Thereby, the status of an intracellular signaling pathway can not be easily evaluated by simply using a gene quantification.
  10. The authors stated that the abscopal effect is related to the RT-induced activation of tumor-specific CD8+ T cells. Are there any data to support CD8+ T cells were activated by TR in this animal model? Besides, decrease of CD4+/CD8+ cell ratio may only be caused by decreasing CD4+ cells or increasing CD8+ cells, thus statistical results in individual counts of CD4+ cells and CD8+ cells should be provided.
  11. Previous study indicated that radiotherapy upregulated PD-L1 through activating EGFR signaling pathway (EBioMedicine. 2018 Feb;28:105-113). However, EGFR and PD-L1 levels were suppressed by TR in the present study (Fig 1A and 5D). Please discuss these contradictory conclusions.
  12. Please confirm TNuF-induced regulations of intracellular signaling molecules and growth inhibition in LLC cells under the intervention of TNuF or its potential active components.

Minor comments:

  1. Abbreviations should be defined at first mention. For example, abbreviations for groups and hematologic parameters.
  2. Superscripts and subscripts of symbols or texts should be carefully checked throughout manuscript. For example, mm3 and 1,25(OH)2 vitamin D3.
  3. Line 100. It should be table 1 instead of table 2.
  4. In Figure 1. Figure legend described that “(E) Representative H&E 115 staining images of lung metastasis tumors (✽). Scale bar = 200 μ (F) Immunofluorescence analysis for Ki-67 in inoculated tumor tissues. Scale bar = 20 μm”, “***, p < 0.001 as compared to the T group”, and “###, p < 0.001 as compared between two groups”. Maybe some data are missed.
  5. Line 143. Maybe, “HSP70” is a typo.
  6. Line 167. Maybe, “HSP90” is a typo.
  7. It’s unclear how to isolate lung metastatic tumor for examining mRNA level of PD-L1? Please add related description in the paragraph of Materials and Methods.
  8. The authors described that TNuF powder was prescribed at the dose of 1 g in 200 μL normal saline. This dose is equal to 5 g/ml or 500% (w/v). Please confirm it. In addition, is the daily dose of mice calculated based on body weight?
  9. It’s unclear where the WAT tissues were collected from?
  10. Is TNuF a commercially available product or an exclusive product designed for this research team? I can not find any information from the website of the manufacturer. Please provide its official name or catalog number. Is TNuF composed of only the components shown in Table 2?
  11. Please provide the composition of blocking buffer solution and staining buffer solution used in immunofluorescence assay (Lines 378-379).
  12. It’s unclear the purposes of anti-CD31 antibody and the primer pair of PD-1 (Lines 377 and 404).

Author Response

Reply to Reviewer 1:
Major comments:
1. The authors have conducted a similar research previously to evaluate the combination effect of TNuF and radiotherapy as well as their possible mechanisms using same animal model and materials (Nutrients. 2019, 11(8): 1944). The results of this published article indicated that TNuF can inhibit tumor metastasis and to enhance radiotherapy effect through activating apoptosis pathway (casapase-3 and Bcl-2) and suppressing the expressions of tumor progression-associated molecules (EGFR, VEGF and HIF-α). Thereby, the novelty is less in the present study, and more advanced data should be provided to substantially improve the quality of this study.
Reply: Regarding the reviewer’s concern our replies are shown as follows:
(1) In the previous research, we had found during the concurrent treatment together with radiotherapy, TNuF dose promote the anticancer effect of radiotherapy. Meanwhile, TNuF also modulates the pathways of apoptosis and angiogenesis in tumor cells. These findings suggest that intake of our total nutrition formula can help cancer patients who are receiving anticancer treatment. However, in the real world, many patients may have used these nutritional supplements before the diagnosis of cancer or before receiving their anticancer treatment, but the influence of nutrition supplements is rarely studied and/or discussed. Therefore, we designed this experiment to see how before contracting cancer the intake of nutrition supplements (TNuF) influences the cancer progression and the efficacy of anticancer radiotherapy (PTN and PTRN groups). We also compared the groups of mice receiving TNuF after tumor inoculation or radiotherapy (TN and TRN groups) as we described in Nutrients. 2019, 11(8): 1944. In this experiment, we found before cancer develops the early intake of TNuF can not only inhibit tumor growth in the PTN group but also enhance the efficacy of radiotherapy in the PTRN group. These results are comparable between the TN and TRN groups, thus implying that cancer patients who have the history of nutrition supplements intake will have a better response to anticancer therapy. We considered this is novel in difference to our previous research.
(2) PD-1/PD-L1 pathway has been recognized as an important route for cancer cells to escape from immune surveillance and suppress the antitumor immune reaction (Noguchi, T. et al. Temporally Distinct PD-L1 Expression by Tumor and Host Cells Contributes to Immune Escape. Cancer Immunol Res; 5(2) February 2017). The interaction of nutrition supplements and tumoral expression of PD-L1 is not fully understood yet. Therefore, we assessed the impact of TNuF on the tumoral PD-L1 expression in this experiment. We found the tumoral expression of PD-L1 can be downregulated with the TNuF supplement irrespective of before or after tumor inoculation. We think this is the second novelty in this report.
(3) Although we have confirmed that TNuF can downregulate the expressions of apoptosis (casapase-3 and Bcl-2), proliferation (EGFR), and angiogenesis (VEGF and HIF-α) in tumor cells, its effect on other pro-tumoral markers has not yet been evaluated. STAT3 is closely related to the cell proliferation induced by EGFR activation (Harada, D. et al. The Role of STAT3 in Non-Small Cell Lung Cancer. Cancers; 2014, 6, 708-722). The proliferation marker Ki-67 is in poor response to anticancer treatment and poor outcome in lung cancer patients (Warth, A. et al. Tumour cell proliferation (Ki-67) in non-small cell lung cancer: a critical reappraisal of its prognostic role. Br J Cancer. 2014 Sep 9; 111(6): 1222–1229). AXL is an angiogenetic marker in association with the invasion and progression of lung cancer (Shieh, Y-S. et al. Expression of Axl in Lung Adenocarcinoma and Correlation with Tumor Progression. Neoplasia. Vol. 7, No. 12, December 2005, pp. 1058 – 1064). TNF-alpha is a pro-inflammatory cytokine, and its overexpression is related to sarcopenia and cachexia in lung cancer patients (Icard, P. et al. Sarcopenia in resected non-small cell lung cancer: let’s move to patient-directed strategies. J Thorac Dis 2018;10(Suppl 26):S3138-S3142). By taking these cues, we also detected these markers in this study for detailing tumor proliferative, angiogenetic, and sarcopenia pathways. We think this is another novelty in this report.

2. The authors indicated that the single treatment with RT or TNuF did not significantly affect the growth of tumors (Lines 88-89). However, all groups except T group have been labeled with a statistical asterisk at day 21 (Fig 1Β).
Reply: Thank you for pointing out this error. We have corrected this error and highlighted the correction in red color.
3. The authors stated that RT will result in sarcopenia. However, the intervention of TR does not change lean BW as well as it can largely increase sartorius M as compared to that of the T group (Table 1). Is this result implied that the animal tumor model used in this study cannot truly reflect clinical findings or phenomenon? Moreover, the authors also stated that TNuF can improve RT-induced sarcopenia. However, no significant difference has been found between TR and intervention groups of TNuF (PTRN and TRN; Table 1).
Reply: For this concern, our results showed that the concurrent TNuF treatment can help RT to reduce cancer (not RT) with sarcopenia through its inhibition to TNF-alpha. We have corrected the corresponding content and highlighted the changes in red color.
4. No description regarding hematologic parameters (Table 1) has been mentioned in the paragraph of Results.
Reply: Thank you for bringing this missing point to our attention. We have added this information in the text highlighted in red color.
5. The authors stated that combination of TNuF and RT has synergistic effect in many regulations. However, are there any results to differentiate or evidence this phenomenon is a synergistic or additive effect. Maybe, “combination index” should be examined.
Reply: For this concern, we agree that TNuF may provide an additive but not synergistic effect on RT. We have rephrased the sentence that is highlighted in red color. The “combination index” suggestion will certainly be examined in our next study.
6. All pictures have poor resolution, and their size are too small. It’s difficult to recognize or extract any useful information from them.
Reply: High-resolution pdf files have been made.
7. Main purpose of immunohistostaining analysis is to understand the locations or distributions of detected proteins within tissue sections. This technique is not suitable for quantifying expression levels of detected proteins. One of reasons is detected proteins are usually heterogeneously distributed within a tissue. Thereby, it’s hard to estimate the overall level of a protein within a tissue based on a small amount of data obtained from some tissue sections. Maybe, Western blot or RT-qPCR would seem a more appropriate choice to quantify expression levels of detected molecules.
Reply: Thank you for the suggestions. We took the cue using immunohistostaining from previous reports that were considered in a position to provide a general trend for the current format. We will use the IHC stating for a qualitative analysis and perform WB or RT-qPCR for a quantitative analysis in next study.
8. How do the authors confirmed that detected signals are distributed in tumor cells, vascular cells or extracellular matrix on tissue sections?
Reply: Before IF staining, we performed H&E staining to find out the distribution of tumor cells, immune cells, and vascular cells on tissue sections. High expressions of Ki-67 and EGFR in LLC cells have already been reported in several papers (Wang, C. et al. Oral 4-(N)-stearoyl gemcitabine nanoparticles inhibit tumor growth in mouse models. Oncotarget; 8(52), September 2017, and Lai, M-D. et al. The effects of DNA formulation and administration route on cancer therapeutic efficacy with xenogenic EGFR DNA vaccine in a lung cancer animal model. Genetic Vaccines and Therapy. 2009, 7:2). HIF-alpha is usually expressed in tumor cells in the lung cancer tissue (Karetsi, E. et al. Differential expression of hypoxia-inducible factor 1a in non-small cell lung cancer and small cell lung cancer. CLINICS. 2012;67(12):1373-1378). CD4 and CD8 are known markers of T helper and cytotoxic T cells, but there is no known information about CD4 or CD8 expression in Lewis Lung Carcinoma cells.
9. Proteins play critical effectors in transductions and activations of intracellular signaling pathways, whose processes not only involved in change of expression level but also post-translational regulations (e.g. protein phosphorylation, degradation from latent form to active form) and translocations. Thereby, the status of an intracellular signaling pathway can not be easily evaluated by simply using a gene quantification.
Reply: Thank you for the suggestions. We agree that only one simple detection of expression of genes or proteins is insufficient to answer the whole state or function of any intracellular signaling pathway. We realize that any finding of gene quantification in this article should be viewed as a “clue” that there is an influence to the specific intracellular pathways. For further understanding, we will exam the change of post-translational regulations of proteins as suggested in our future works.
10. The authors stated that the abscopal effect is related to the RT-induced activation of tumor- specific CD8+ T cells. Are there any data to support CD8+ T cells were activated by TR in this animal model? Besides, decrease of CD4+/CD8+ cell ratio may only be caused by decreasing CD4+ cells or increasing CD8+ cells, thus statistical results in individual counts of CD4+ cells and CD8+ cells should be provided.
Reply: In the papers of Yokouchi, H. et al. (Anti‐OX40 monoclonal antibody therapy in combination with radiotherapy results in therapeutic antitumor immunity to murine lung cancer. Cancer Sci. 2008 Feb;99(2):361-7) and Chamoto, K. et al. (Combination immunotherapy with radiation and CpG‐based tumor vaccination for the eradication of radio‐ and immuno‐resistant lung carcinoma cells. Cancer Sci. 2009 Apr;100(5)), they found that tumor-specific CD8+ T cells were increased in LLC-bearing mice receiving radiotherapy. The data are consistent with our results that CD8+ T cells can be activated by radiotherapy in the animal model. We used the immunofluorescence analysis to detect the presentation of CD4+/CD8+ cells in tissue section, thus defining the ratio of CD4+/CD8+ cells by measuring the fluorescence intensity. However, we agree that this method is not as accurate as the direct calculation of cell numbers by flow cytometry. In the future works, we will use flow cytometry to calculate the cell numbers of CD4+/CD8+ T cells.
11. Previous study indicated that radiotherapy upregulated PD-L1 through activating EGFR signaling pathway (EBioMedicine. 2018 Feb;28:105-113). However, EGFR and PD-L1 levels were suppressed by TR in the present study (Fig 1A and 5D). Please discuss these contradictory conclusions.
Reply: According to the papers of Song, X. et al. (Radiotherapy Upregulates Programmed Death Ligand-1 through the Pathways Downstream of Epidermal Growth Factor Receptor in Glioma. EBioMedicine. 2018 Feb;28:105-113) and Concha-Benavente, F. et al. (Identification of the cell-intrinsic and -extrinsic pathways downstream of EGFR and IFNgamma that induce PD-L1 expression in head and neck cancer. Cancer Res. 76, 1031–1043), the upregulation of PD-L1 in tumor cells is correlated to two mechanisms — the “extrinsic” pathway which depends on interferon gamma produced by NK and CD8+ T cells, and the “intrinsic” pathway that relies on the EGFR/JAK2 signaling pathway within glioma or head and neck cancer cells. Therefore, we think the contradictory conclusion in our article may be attributed to (1) different cell lines (Lewis Lung Carcinoma vs glioma or head and neck cancer), (2) different study designs (animal vs cell-line or clinical), and (3) different protocols of RT treatment between ours and Song, et al. We are keen to investigate the EGFR and PD-L1 expressions in glioma or head and neck cancer cells using our animal model in the future study.
12. Please confirm TNuF-induced regulations of intracellular signaling molecules and growth inhibition in LLC cells under the intervention of TNuF or its potential active components.
Reply: In our previous studies, we reported that selenium and fish oil, two major components of TNuF, can inhibit the growth of Line 1 lung cancer cell and regulate the intracellular signaling molecules of inflammation, angiogenesis, and apoptosis in both EMT6 breast and Line 1 lung cancers (Guo, C-H. et al. Distribution of Selenium and Oxidative Stress in Breast Tumor-Bearing Mice. Nutrients 2013, 5, 594-607, Wang, H. et al. Reduction of Splenic Immunosuppressive Cells and Enhancement of Anti-Tumor Immunity by Synergy of Fish Oil and Selenium Yeast. PLoS ONE; 2012. 8(1): e52912, and Wang, H. et al. Skeletal muscle atrophy is attenuated in tumor-bearing mice under chemotherapy by treatment with fish oil and selenium. Oncotarget; 2015. Vol. 6, No. 10). In the animal and clinical studies, we also discovered that TNuF can inhibit the growth of murine LLC cells and stabilize body weight in patients with head and neck cancer on receiving radiotherapy (Liu, Y-M. et al. Antitumor, Inhibition of Metastasis and Radiosensitizing Effects of Total Nutrition Formula on Lewis Tumor-Bearing Mice. Nutrients 2019, 11, 1944, and Yeh, K-Y. et al. Omega-3 fatty acid-, micronutrient-, and probiotic-enriched nutrition helps body weight stabilization in head and neck cancer cachexia. Oral Surg Oral Med Oral Pathol Oral Radiol 2013;116:41-48). Added together, we propose that TNuF can regulate intracellular signaling molecules and growth inhibition in LLC cells.
Minor comments:
1. Abbreviations should be defined at first mention. For example, abbreviations for groups and hematologic parameters.
Reply: We have corrected these errors and highlighted the corrections in red color.
2. Superscripts and subscripts of symbols or texts should be carefully checked throughout manuscript. For example, mm3 and 1,25(OH)2 vitamin D3.
Reply: We have corrected these errors and highlighted the corrections in red color.
3. Line100. It should be table 1 instead of table 2.
Reply: We have corrected these errors and highlighted the corrections in red color.
4. In Figure 1. Figure legend described that “(E) Representative H&E 115 staining images of lung metastasis tumors (✽). Scale bar = 200 μ (F) Immunofluorescence analysis for Ki-67 in inoculated tumor tissues. Scale bar = 20 μm”, “***, p < 0.001 as compared to the T group”, and “###, p < 0.001 as compared between two groups”. Maybe some data are missed.
Reply: We have deleted these two unrelated sentences.
5. Line143. Maybe, “HSP70” is a typo.
Reply: Heat shock protein 90 is right. We have corrected these errors and highlighted the corrections in red color.
6. Line 167. Maybe, “HSP90” is a typo.
Reply: Heat shock protein 70 is right. We have corrected these errors and highlighted the corrections in red color.
7. It’s unclear how to isolate lung metastatic tumor for examining mRNA level of PD-L1? Please add related description in the paragraph of Materials and Methods.
Reply: The condition of lung metastasis was inspected visually after sacrifice, and the visible tumors were dissected for the following experiments. We have added the description in Materials and Methods.
8. The authors described that TNuF powder was prescribed at the dose of 1 g in 200 μL normal saline. This dose is equal to 5 g/ml or 500% (w/v). Please confirm it. In addition, is the daily dose of mice calculated based on body weight?
Reply: The correct dose is 1 g in 5 ml normal saline per day, equal to 0.2 g/ml or 20% (W/V). We have corrected this error and highlighted them in red color. Meanwhile, the daily dose of every mice was fixed and did not altered by body weight.
9. It’s unclear where the WAT tissues were collected from?
Reply: The white adipose tissues were collected from the gonadal organs in mice. We have added the description in the Materials and Methods and highlighted it in red color.
10. Is TNuF a commercially available product or an exclusive product designed for this research team? I can not find any information from the website of the manufacturer. Please provide its official name or catalog number. Is TNuF composed of only the components shown in Table 2?
Reply: TNuF is a commercially available product produced by DoWell Laboratories in Tustin, CA (https://www.dowell-lab.com/store/p10/NUTRAWELL_%28450_g%29.html#/). The official name is NUTRAWELL. Table 2 lists the major components of TNuF.
11. Please provide the composition of blocking buffer solution and staining buffer solution used in immunofluorescence assay (Lines 378-379).
Reply: The blocking buffer solution contained 1X PBS, 5% normal serum, and 0.3% Triton X-100. The antibody dilution buffer solution contained 1X PBS, 1% BSA, and 0.3% Triton X-100. We have added these descriptions in the paragraph of Materials and Methods.
12. It’s unclear the purposes of anti-CD31 antibody and the primer pair of PD-1 (Lines 377 and 404).
Reply: We are sorry for the unnecessary description of anti-CD31 and PD-1 primers. We have removed these sentences from Materials and Methods.

Reviewer 2 Report

In the manuscript, the authors present an experimental study regarding the effects of the preventive nutrition supplement in an experimental model of lung cancer in mice. They included in the study 42 mice that were divided in 7 groups. As a nutrition supplement, the authors used TNuF.  TNuF was administrated in mice before or after tumor inoculation. In my opinion it is an interesting manuscript that can be published. The conclusions of the manuscript are pertinent. Also, I have some observations :

  1. There are some small grammar and spelling errors in English. The English language must be revised. 
  2. There are differences between the conclusions part of the main manuscript and the conclusions part of the abstract. Please do not include in the conclusions part of the manuscript data regarding TNuF
  3. Please include at the end of the discussions part of the main manuscript some data regarding the limitations of the study

Author Response

Reply to Reviewer 2:

In the manuscript, the authors present an experimental study regarding the effects of the preventive nutrition supplement in an experimental model of lung cancer in mice. They included in the study 42 mice that were divided in 7 groups. As a nutrition supplement, the authors used TNuF. TNuF was administrated in mice before or after tumor inoculation. In my opinion it is an interesting manuscript that can be published. The conclusions of the manuscript are pertinent. Also, I have some observations:

1. There are some small grammar and spelling errors in English. The English language must be revised.

Reply: Thank you for the suggestions. We have revised our article and corrected these errors.

2. There are differences between the conclusions part of the main manuscript and the conclusions part of the abstract. Please do not include in the conclusions part of the manuscript data regarding TNuF.

Reply: We have revised our conclusions following the suggestions.

3. Please include at the end of the discussions part of the main manuscript some data regarding the limitations of the study.

Reply: We have improved the discussion in our article and highlighted them in red color.

Round 2

Reviewer 1 Report

  1. Major comments:

    1. No convincing data or further experiments for responding the reviewer’s major concerns have been provided in the revised manuscript.
    2. Please integrate the responses or explanations for reviewer’s concerns or the controversial results into Discussion section.
    3. The resolutions of numerous pictures (Figs. 2A, 2E, 4A, 5A, 5B and 5C) are so terrible. The resolution of a mosaic can not be substantially improved by enlarging its size. According to the current resolution of these pictures, it is difficult to distinguish the credibility and authenticity of statistical data. Please seek expert assistance to provide high-resolution pictures. In addition, please mention the total numbers of tissue sections of each group used for statistical analysis.
    4. The authors indicated that TNuF is a commercially available product produced by DoWell Laboratories. Simon Hsia, one of the authors in this study, is the founder of DoWell Laboratories, and New Health Enterprise, Inc., the provider of TNuF, maybe a direct selling company. These relationships should be disclosed in the Conflicts of Interest section. In addition, both the provider and producer of TNuF as well as its official name should be mentioned clearly in Materials and Methods section.
    5. The authors indicated that the combination treatment provided a better inhibition to tumor growth than either one of the treatments as compared to the single treatment with RT or TNuF (Lines 91 to 92). Does the single treatment with TNuF mean the PTN and the TN groups? However, only the T group was used as the control group to perform statistical analysis based on the figure legend of figure 1. It’s unclear what basis the authors affirm this statement?
    6. The authors stated that these results suggested that TNuF can synergize with RT to improve cancer-induced sarcopenia (Lines 128 to 129). However, TR alone can significant reverse cancer-induced sarcopenia as compared to the T group (Table 1). Is there any statistical difference between the TR group and the combination groups (PTRN or TRN group)?
    7. No description regarding experimental results of hematologic parameters (Table 1) were mentioned in the paragraph of Results.
    8. The combination of TR and TNuF has synergistic effect still be described in some sentences (Lines 250 and 332) even if the authors can not differentiate whether this activity is additive or synergistic effect in the present study.
    9. The authors indicated the distribution of tumor cells, immune cells, and vascular cells on tissue sections using HE staining. However, it’s unclear how to identify different cells by simply using a HE staining rather than an immunohistochemical staining. In addition, please provide these staining data as supplementary materials.
    10. Please provide individual fluorescent intensities of CD4+ and CD8+ cells on tissue sections as supplementary data to confirm abscopal effect is related to the RT-induced activation of tumor- specific CD8+ T cells in this animal model.
    11. The authors proposed that selenium and fish oil, two major components of TNuF, maybe involved in TNuF-induced regulations of intracellular signaling molecules and growth inhibition in LLC cells based on the researches of different cancer cells or types. As mentioned by the authors to explain controversial effect of RT on EGFR and PD-L1 levels, different cells maybe have different responses to same experimental condition. Thereby, please confirm the regulations of partial (or critical) intracellular signaling molecules and growth inhibition in LLC cells treated with potential active components, selenium and fish oil.

    Minor comments:

    1. Abbreviations should be defined “at first mention”. For example, PTN and TN do not be defined even if these abbreviations were appeared in figure 1 (Lines 1 to 121).

Author Response

Reply to Reviewer 1:

Major comments:

1. No convincing data or further experiments for responding the reviewer’s major concerns have been provided in the revised manuscript.

Reply: In our previous paper published in Nutrients 2019, 11(8): 1944, we reported that the combination treatment of TNuF and RT after tumor inoculation can improve anticancer response and cachectic symptoms in lung cancer-bearing mice by upregulating tumoral casapase-3 and Bcl-2 signaling, but downregulating EGFR, VEGF and HIF-α pathways in tumors. However, on one ever reported what would result on the tumor growth and/or the response of RT if the TNuF is prescribed before tumor inoculation. Besides, there is no repot about the influence of TNuF on other cancer-related signaling proteins like PTEN (an inhibitor of PIP3 in EGFR pathway), STAT3 (cancer proliferation and inflammation), Ki-67 (mitosis and proliferation), AXL (angiogenesis), PD-L1 (immunosuppression), TNF-α (inflammation and sarcopenia), and HSP70/90 (chaperon proteins) in lung cancer-bearing mice. Therefore, in this experiment we not only investigated the effect of early TNuF prescription before tumor inoculation to cancer growth and anticancer response of RT, but we also examined the expressions of STAT3, Ki-67, AXL, PD-L1, and TNF-α for a more detailed comprehension to the effect of TNuF to cancer associated signaling pathways.

In this experiment, we divided mice into six groups: mice of T group only received tumor inoculation, mice of TR group received RT after tumor inoculation, mice of PTN group started to receive the TNuF supplement before tumor inoculation, mice of PTRN group began to receive TNuF before tumor inoculation and then receive RT after tumor inoculation, mice of TN group received TNuF after tumor inoculation, and mice of TRN group received the combination treatment of TNuF and RT after tumor inoculation. We found the early prescription of TNuF before tumor inoculation significantly suppressed the growth of tumor [mean tumor volume on Day 21 (mm3): 2940 in T group, 2011 in PTN group, 497 in PTRN group, p < 0.05], the anticancer efficacy of RT [mean tumor weight on Day 21 (g): 1 in T group, 0.47 in TR group, 0.16 in PTRN group, p < 0.05]. In the molecular analysis, we found that no matter to the timing of prescription, a single or combination treatment, TNuF significantly upregulates PTEN [relative level: 1 in T group, 0.14 in TR group, 0.47 in PTN group, 0.22 in PTRN group, 0.28 in TN group, 0.4 in TRN group, p < 0.05], AXL [relative level: 1 in T group, 2.3 in TR group, 3.9 in PTN group, 0.11 in PTRN group, 0.28 in TN group, 0.11 in TRN group, p < 0.05]. In contrast, TNuF also downregulates STAT3 [relative level: 1 in T group, 0.4 in TR group, 0.5 in PTN group, 0.3 in PTRN group, 2.6 in TN group, 2.8 in TRN group, p < 0.01] and Ki-67 [decreased fluorescent intensity in the groups of TNuF and/or RT as compared to T group, Figure 2E] in tumors. The results of other EGFR and VEGF signaling proteins were similar to those in our Nutrients article. These results indicated that the early nutrition intake before cancer diagnosis still can help to inhibit the growth of tumor and to promote the anticancer response of RT via modulating tumoral apoptotic, proliferative, and angiogenic pathways. We also consider these findings provide a more detailed comprehension to the interaction between nutrition, cancer cells and RT; a preliminary answer to the effect of previous nutrient intake to cancer patients in the future; and hence, there are novelties as compared to our previous research published in Nutrients.

The PD-1/PD-L1 pathway is important to cancer cells to escape immune surveillance and anticancer immune reaction in the hosts. An elevating CD4/CD8 cell ratio in tumor tissue is also related to cancer progression. However, the relationship between tumoral PD-1/PD-L1 pathways, CD4/CD8 ratio, and nutrients is not fully understood. Hence, we investigated the tumoral PD-L1 expression and CD4/CD8 cell ratio in this experiment. We noted that tumoral expression of PD-L1 was significantly suppressed by TNuF [relative level: 1 in T group, 0.22 in TR group, 0.12 in PTN group, 0.26 in PTRN group, 0.28 in TN group, 0.22 in TRN group, p < 0.05]. Meanwhile, comparing to T and TR groups, the fluorescent intensity of CD4 cells was decreased and that of CD8 cells was enhanced in tumor tissue after TNuF treatment, which transferred to a decreasing CD4/CD8 ratio (Figure 5B). These results indicated that TNuF can downregulate PD-L1 expression in tumor cells and promote the infiltration of anticancer cytotoxic T cells in tumor tissue. These findings also suggested that the nutrient intake can strengthen anticancer immunity in cancer patients, without regard to the timing of prescription. We consider this discovery is another novelty comparing to our previous research in Nutrients.

2. Please integrate the responses or explanations for reviewer’s concerns or the controversial results into Discussion section.

Reply: We’ve integrated our explanations with regard to reviewer’s concerns into Introduction and Discussion sections and highlighted in red color.

3. The resolutions of numerous pictures (Figs. 2A, 2E, 4A, 5A, 5B and 5C) are so terrible. The resolution of a mosaic can not be substantially improved by enlarging its size. According to the current resolution of these pictures, it is difficult to distinguish the credibility and authenticity of statistical data. Please seek expert assistance to provide high-resolution pictures. In addition, please mention the total numbers of tissue sections of each group used for statistical analysis.

Reply: Because of the size limitation in the submission system, we can not upload pictures of higher resolution as recommended. We will directly e-mail the pdf files of high-resolution figures to editors who will forward to reviewers. Because of the limitation of our equipment, we are not able to take high-resolution IF pictures for this experiment. We will seek assistance outside our institute in future experiments. Nevertheless, we have selected six tissue sections of each group for IF and statistical analysis. We’ve added corresponding descriptions into the figure legends and highlighted in red color.

4. The authors indicated that TNuF is a commercially available product produced by DoWell Laboratories. Simon Hsia, one of the authors in this study, is the founder of DoWell Laboratories, and New Health Enterprise, Inc., the provider of TNuF, maybe a direct selling company. These relationships should be disclosed in the Conflicts of Interest section. In addition, both the provider and producer of TNuF as well as its official name should be mentioned clearly in Materials and Methods section.

Reply: We have added a relevant description in Materials and Methods as well as Conflicts of Interest sections.

5. The authors indicated that the combination treatment provided a better inhibition to tumor growth than either one of the treatments as compared to the single treatment with RT or TNuF (Lines 91 to 92). Does the single treatment with TNuF mean the PTN and the TN groups? However, only the T group was used as the control group to perform statistical analysis based on the figure legend of figure 1. It’s unclear what basis the authors affirm this statement?

Reply: The correct sentence is “As compared to the single treatment with RT or TNuF, the combination treatment of TNuF and RT (PTRN and TRN groups) had a trend to provide a better inhibition to tumor growth”. Meanwhile, in the figure legend, we have added the description “#, p < 0.05 as compared to the TR group. ##, p < 0.01 as compared to the TR group”. We have revised the relative description and highlighted in red color.

6. The authors stated that these results suggested that TNuF can synergize with RT to improve cancer-induced sarcopenia (Lines 128 to 129). However, TR alone can significant reverse cancer-induced sarcopenia as compared to the T group (Table 1). Is there any statistical difference between the TR group and the combination groups (PTRN or TRN group)?

Reply: The correct sentence is “TNuF and RT both can improve cancer-induced sarcopenia”. We have revised the description and highlighted in red color.

7. No description regarding experimental results of hematologic parameters (Table 1) were mentioned in the paragraph of Results.

Reply: We’ve added a relevant description in the Results section highlighted in red color.

8. The combination of TR and TNuF has synergistic effect still be described in some sentences (Lines 250 and 332) even if the authors can not differentiate whether this activity is additive or synergistic effect in the present study.

Reply: We’ve corrected these errors, revised our description, and highlighted in red color.

9. The authors indicated the distribution of tumor cells, immune cells, and vascular cells on tissue sections using HE staining. However, it’s unclear how to identify different cells by simply using a HE staining rather than an immunohistochemical staining. In addition, please provide these staining data as supplementary materials.

Reply: We have added the H&E staining photos in supplementary materials.

10. Please provide individual fluorescent intensities of CD4+ and CD8+ cells on tissue sections as supplementary data to confirm abscopal effect is related to the RT- induced activation of tumor- specific CD8+ T cells in this animal model.

Reply: The individual fluorescent intensities of CD4+ and CD8+ cells on tissue sections have been added in supplementary materials.

11. The authors proposed that selenium and fish oil, two major components of TNuF, maybe involved in TNuF-induced regulations of intracellular signaling molecules and growth inhibition in LLC cells based on the researches of different cancer cells or types. As mentioned by the authors to explain controversial effect of RT on EGFR and PD-L1 levels, different cells maybe have different responses to same experimental condition. Thereby, please confirm the regulations of partial (or critical) intracellular signaling molecules and growth inhibition in LLC cells treated with potential active components, selenium and fish oil.

Reply: Schley, P.D. et al. have found the fish oil can decrease the phosphorylation of both EGFR and p38 MAPK in cancer cells (J. Nutr. 137: 548–553, 2007). Chen Y.-C. et al. also noticed that selenite, a selenium containing protein, can decrease EGFR expression in cancer cells (Nutrients 2013, 5, 1149-1168). Meanwhile, Nair D. et al. have reported that methylselenic acid and selenite can decrease PD-L1 expression in cancer cells (Front. Oncol. 2018, 8, 407). Reported by Liao, C.-H. et al., the combination of selenium and fish oil was able to reverse the acquired resistance of EGFR tyrosine kinase inhibitors in cancer cells (Mar. Drugs 2020, 18, 399). Therefore, we proposed that selenium and fish oil may involve in TNuF-induced regulation of signaling pathways and inhibition in LLC cells. The research of intracellular signaling and molecular mechanism is beyond the focus of this experiment. But in the future experiment, we will continue to study the regulations of partial (or critical) intracellular signaling molecules and growth inhibition in LLC cells treated with selenium and fish oil. Thank you very much!

Minor comments:

1. Abbreviations should be defined “at first mention”. For example, PTN and TN do not be defined even if these abbreviations were appeared in figure 1 (Lines 1 to 121).

Reply: We have added the definition of each group including PTN and TN in Result 2.1 section and highlighted in red color.

Round 3

Reviewer 1 Report

Major comments:

  1. Picture resolutions of the tissue sections are still poor, and no any original files were uploaded as supplementary materials that can help to valid the correctness of these data. Thereby, it's hard to distinguish or obtain any information regarding cell sources of these fluorescent signals.
  2. The authors said that the H&E staining photos have been provided as supplementary materials to explain how to distinguish the distribution of tumor cells, immune cells, and vascular cells on tissue sections. However, only a excel file regarding data of CD4+ and CD8+ levels were uploaded.
  3. The authors described that data were analyzed by multivariate ANOVA test. In fact, difference of CD4+/CD8+ ratio among all group is analyzed using the Student’s t-test according to the content of the supplementary file. Thereby, the correctness of all experimental description and statistical methods used in the present study should be comprehensive and rigorous confirmed.
  4. The present study showed that RT can improve cancer-induced sarcopenia. However, it has also be described that RT also brings patients lots of adverse effects such as muscle wasting. Please add a discussion to explain this controversial effect.
  5. Please add a discussion to explain why TNuF and its bioactive components exhibited cell-specific toxicity on cancer cells rather than normal cells.
  6. It’s an over conclusion that TNuF takes anti-angiogenetic and immunomodulatory effects on lung cancer cells (Lines 436-437). We cannot infer that these physiological phenomena are suppressed only by relying on the changes of expression levels of some molecules involved in angiogenesis or immunomodulation. Maybe more data from cell-based assay or animal model should be included to support these inferences. Besides, a molecule can be regulated or cross talked by different upstream signaling pathways, thus it cannot be stated that a pathway has been blocked or regulated by simply detecting one of molecules involved in a signaling pathway. 

Minor comments:

  1. Panel label “A” missed on figure 4.

Author Response

Reply to Reviewer 1:

Major comments:

1. Picture resolutions of the tissue sections are still poor, and no any original files were uploaded as supplementary materials that can help to valid the correctness of these data. Thereby, it's hard to distinguish or obtain any information regarding cell sources of these fluorescent signals.

Reply: We are so regretful to hear the recurrence of the concern poor picture resolution. We think the problem may be attributed to the submission system or miscommunication between reviewer and editor. We have indeed uploaded original photos and figures as separate files for the reviewer to discern the correctness of the data presented in the main text. To avoid this nuisance happening again, we will send corresponding copies parallelly to editor while submitting this new version and will remind the editor forwarding them directly to reviewer #1. We will also like to provide the Google Drive Sharable Link of our supplement files as follows. https://drive.google.com/drive/folders/18BsKokAIgY_i6G2w728Q-XhwfaVS7Sl-?usp=sharing

2. The authors said that the H&E staining photos have been provided as supplementary materials to explain how to distinguish the distribution of tumor cells, immune cells, and vascular cells on tissue sections. However, only a excel file regarding data of CD4+ and CD8+ levels were uploaded.

Reply: Please refer to the reply for issue 1.

3. The authors described that data were analyzed by multivariate ANOVA test. In fact, difference of CD4+/CD8+ ratio among all group is analyzed using the Student’s t-test according to the content of the supplementary file. Thereby, the correctness of all experimental description and statistical methods used in the present study should be comprehensive and rigorous confirmed.

Reply: We thank reviewer #1 for bringing up this mistake to our attention. We have corrected it in Method.

4. The present study showed that RT can improve cancer-induced sarcopenia. However, it has also be described that RT also brings patients lots of adverse effects such as muscle wasting. Please add a discussion to explain this controversial effect.

Reply: We thank reviewer #1 for this suggestion. Although RT has been reported to induce some adverse effects such as muscle wasting in patients of lung cancer [1], RT, however, was shown to improve cancer-induced sarcopenia in this experiment. We propose this controversial effect can be attributed to a short course of RT that turns out to be a good anticancer response to LLC cells. We’ve added this description and highlighted in red in Discussion.

5. Please add a discussion to explain why TNuF and its bioactive components exhibited cell-specific toxicity on cancer cells rather than normal cells.

Reply: We are glad to hear this and thank reviewer #1 for this suggestion. Husbeck, et al. and Fico, et al. reported that selenium administration can induce selective cytotoxicity in adenocarcinoma cell lines, but not in normal cells [2,3]. Additionally, n-3 PUFA has been proved to cause selective cytotoxicity against cancer cell lines but no significant toxicity against normal cells [4]. These findings indicate that selenium and n-3 PUFA, two major components of TNuF, exhibit cell-type specific toxicity on cancer cells rather than normal cells. We’ve added this description and highlighted in red in Discussion.

6. It’s an over conclusion that TNuF takes anti-angiogenetic and immunomodulatory effects on lung cancer cells (Lines 436-437). We cannot infer that these physiological phenomena are suppressed only by relying on the changes of expression levels of some molecules involved in angiogenesis or immunomodulation. Maybe more data from cell-based assay or animal model should be included to support these inferences. Besides, a molecule can be regulated or cross talked by different upstream signaling pathways, thus it cannot be stated that a pathway has been blocked or regulated by simply detecting one of molecules involved in a signaling pathway.

Reply: We thank reviewer #1 for bringing up this suggestion. We’ve corrected this paraph accordingly and highlighted in red in Discussion.

Minor comments:

Panel label “A” missed on figure 4.

Reply: We’ve corrected this error in Figure 4.

References

  1. de van der Schueren, M.A.E.; Laviano, A.; Blanchard, H.; Jourdan, M.; Arends, J.; Baracos, V.E. Systematic review and meta-analysis of the evidence for oral nutritional intervention on nutritional and clinical outcomes during chemo(radio)therapy: current evidence and guidance for design of future trials. Annals of oncology: official journal of the European Society for Medical Oncology 2018, 29, 1141-1153, doi:10.1093/annonc/mdy114.
  2. Fico, M.E.; Poirier, K.A.; Watrach, A.M.; Watrach, M.A.; Milner, J.A. Differential effects of selenium on normal and neoplastic canine mammary cells. Cancer research 1986, 46, 3384-3388.
  3. Husbeck, B.; Nonn, L.; Peehl, D.M.; Knox, S.J. Tumor-selective killing by selenite in patient-matched pairs of normal and malignant prostate cells. Prostate 2006, 66, 218-225, doi:10.1002/pros.20337.
  4. D'Eliseo, D.; Velotti, F. Omega-3 Fatty Acids and Cancer Cell Cytotoxicity: Implications for Multi-Targeted Cancer Therapy. Journal of clinical medicine 2016, 5, doi:10.3390/jcm5020015.
